

# Optimization model for enterprise financial management utilizing genetic algorithms and fuzzy logic

Sujuan Wang[1] and Musadaq Mansoor[2]

[1] School of Accounting, Zhengzhou University of Economics and Business, Zhengzhou, Henan, China
[2] School of Computing Sciences, Pak-Austria Fachhochschule Institute of Applied Sciences and Technology, Haripur, Pakistan

## ABSTRACT

This study explores the complexities of enterprise financial management by optimizing financial models with a particular focus on enhancing risk prediction performance. A multi-objective mathematical model is first developed to establish key optimization goals, including cost reduction, improved capital utilization, and increased economic benefits. This model systematically defines decision variables and optimization objectives, providing a comprehensive framework for enterprise financial management. To improve predictive accuracy, the study integrates genetic algorithms with back-propagation (BP) neural networks, leveraging genetic algorithms to optimize the neural network's parameters and structure. Additionally, a hierarchical reinforcement learning model based on fuzzy reasoning (HRL-FR) is proposed to enhance decision-making capabilities. This model employs hierarchical decision-making and policy optimization, incorporating fuzzy reasoning to address uncertainties in complex and dynamic financial environments. Experimental validation using the Compustat dataset confirms the effectiveness of the proposed model. Key financial variables, including the working capital asset ratio and debt-to-equity ratio, are identified as significant influencers of prediction accuracy, reinforcing the model's robustness. The genetic algorithm's search and optimization process identifies parameter combinations that maximize neural network performance, further improving predictive capabilities. Comprehensive evaluations conducted on the Center for Research in Security Prices (CRSP) and Compustat datasets for 2022 confirm the HRL-FR model's superior ability to predict and analyze enterprise financial management information accurately. The model demonstrates higher profitability, enhanced efficiency, and predictive curves that closely align with optimal financial models. These findings highlight the HRL-FR model's potential as a powerful tool for enterprise financial management optimization, offering valuable insights for risk mitigation and strategic decision-making.

Corresponding author
Sujuan Wang,
wangsujuan_1986@126.com

## INTRODUCTION

In today's fiercely competitive market, enterprises face numerous challenges, particularly financial management. Effective financial management is essential for the survival and growth of enterprises, as it directly influences their profitability, capital liquidity, and risk control. However, traditional financial management methods, often reliant on fixed models and parameters, struggle to adapt to the complexities of dynamic market environments. Consequently, developing innovative financial management optimization models that align with evolving market conditions has become a significant focus in contemporary research.

Computational intelligence techniques, including genetic algorithms and fuzzy logic, offer novel approaches to optimizing financial management (*Nica, Delcea & Chiriță, 2024*). Genetic algorithms (GAs) (*Alhijawi & Awajan, 2024*) are a type of bio-inspired optimization method that mimics the natural processes of selection, crossover, and mutation observed in biological evolution. GAs have been employed in financial management to optimize various objectives, such as investment portfolios, cost control, and risk management. By generating an initial population of potential solutions and evaluating their fitness based on a problem-specific evaluation function, GAs iteratively select the most promising candidates. These chosen individuals undergo crossover and mutation operations, improving solution quality continuously (*Taha, Abdullah & Rashid, 2024*).

In recent years, GAs have made significant strides in financial management, particularly in core areas such as portfolio optimization, cost control, and risk management. By finely tuning asset allocation, GAs have successfully helped enterprises find an ideal balance between risk and return, enhancing their profit potential and risk resistance. However, despite the immense potential demonstrated by GAs in these fields, their decision-making process still appears somewhat mechanical, lacking the necessary intelligent adaptability.

This article addresses this shortcoming by integrating fuzzy logic with genetic algorithms. Fuzzy logic (*Woźniak et al., 2024*), a control method based on human fuzzy thinking, excels at handling uncertainty and incomplete information in complex and uncertain environments. In financial management, fuzzy logic can skillfully construct fuzzy sets using economic indicators and reason through meticulously designed fuzzy rules to derive key outputs such as financial early warnings and investment strategies. This approach not only fully considers the fuzziness and uncertainty among financial indicators but also significantly improves the accuracy and reliability of prediction results.

The existing financial management models exhibit deficiencies when confronted with complex and volatile market conditions. Traditional financial management approaches, often relying on fixed models and parameters, struggle to fully encompass corporate financial management's complexity and diversity. Furthermore, they possess shortcomings in risk prediction performance. This article aims to explore an optimization method combining genetic algorithms and fuzzy logic further to elevate the level of intelligence in enterprise financial management. By deeply analyzing the current application status and limitations of genetic algorithms in financial management, this article proposes an

innovative fusion model to provide more intelligent, efficient, and accurate decision support for enterprise financial management.

The specific contributions of this article are as follows:

(1) Development of a multi-objective data model: Considering the complexity and diversity of enterprise financial management, a multi-objective mathematical model is constructed. This model incorporates objectives such as cost reduction, enhanced fund utilization, and increased economic efficiency, providing a comprehensive reflection of enterprise financial management practices.

(2) Integration of GAs and neural networks: GAs are employed to optimize the parameters and structure of neural networks, thereby improving prediction accuracy and decision-making effectiveness. This approach addresses the limitations of traditional parameter determination methods in identifying optimal parameter combinations.

(3) Construction of a hierarchical reinforcement learning model based on fuzzy inference: By leveraging human *a priori* knowledge, this model achieves end-to-end learning and training, accelerates convergence, enhances decision-making outcomes, and improves prediction accuracy.

The structure of this article is as follows:

"Related Work" reviews the current state of research on GAs and fuzzy logic, analyzing their strengths and weaknesses in the context of enterprise financial management optimization.

"Methodology" introduces the multi-objective optimization model developed in this study, the prediction model combining GAs with back-propagation (BP) neural networks, and the hierarchical reinforcement learning (HRL-FR) model based on fuzzy reasoning.

"Experimental analysis" presents the experimental results, evaluating the performance of the proposed algorithms and comparing them with the existing enterprise management optimization prediction model, LPM. The impact of the multi-objective approach, GAs, BP neural networks, and the HRL-FR model on corporate financial risk prediction is analyzed.

"Conclusion" concludes the article by summarizing the performance of the enterprise financial management optimization model constructed in this study and suggesting directions for future research.

## RELATED WORK

### Genetic algorithms

GA is a heuristic search algorithm inspired by natural selection and genetics principles, widely used to address optimization and search problems. In enterprise financial management optimization, GA can enhance the allocation of resources within financial management systems, thereby improving overall efficiency and reliability. However, traditional GA often suffers from premature convergence, leading to suboptimal solutions

by getting trapped in local optima. Consequently, numerous improvements have been proposed to address this limitation.

For instance, *Alhijawi & Awajan (2024)* introduced enhancements to the selection operator, fitness evaluation, and elite operations, resulting in a GA with significantly stronger search capabilities. *Abdelkhalek et al. (2024)* proposed adaptive adjustments to replication, crossover, and mutation rates and demonstrated through test functions and repeated experiments that the improved GA consistently achieved global optimal solutions. *Wang et al. (2024)* incorporated periodic coding and priority decoding alongside adaptive probability strategies (*Hao et al., 2025*) and multi-swarm joint optimization techniques (*Chen et al., 2025*). These modifications enhanced the accuracy of the GA compared to traditional methods and outperformed the GA combined with time-sensitive networks (TSN) (*Xue et al., 2024*).

Further, *Usvakangas, Tuovinen & Neittaanmäki (2024)* applied methods such as the greedy algorithm PMX (Partially Mapped Crossover) (*Hardi, Manik & Febriana, 2024*), reversal operators, and immigration strategies to low-carbon cold chain logistics studies, effectively addressing the limitations of traditional GA. Similarly, *Mazouzi et al. (2024)* employed adaptive multi-swarm GA to optimize control system integration, significantly improving efficiency, with each test achieving or approaching optimal solutions. *Pan et al. (2024)* focused on energy consumption in network devices, refining mutation strategies in GA. Their results demonstrated improved GA effectively reduces storage space wastage, energy consumption, and latency.

The advantages and disadvantages of the above genetic algorithms are shown in Table 1. Designing an enterprise financial management optimization model involves numerous interdependent and complex factors, including the organization's financial status, business environment, market conditions, and risk tolerance. While the improvements above have enhanced the search and optimization capabilities of GA, challenges persist in achieving optimal parameter selection when addressing such multifaceted factors.

## Fuzzy logic

In the integration and innovation of financial management, artificial intelligence (AI) demonstrates clear advantages in data processing and decision support. It excels in handling large-scale and complex financial data, enabling rapid and precise analysis and predictions, significantly enhancing enterprise financial management's efficiency and accuracy. However, AI systems cannot entirely replace the role of human expertise. This limitation stems from AI decision-making relying on training data and algorithmic models, whereas human expertise is derived from extensive learning and practical experience.

Fuzzy logic control (*Maroua et al., 2024*), a control methodology based on the principles of fuzzy reasoning, applies fuzzy sets and fuzzy rules to the decision-making process. *Precup, Nguyen & Blažič (2024)* highlights its application in managing non-deterministic systems by modeling and addressing uncertainty and fuzziness. Similarly, *Gorgin et al. (2024)* describes how fuzzy logic control maps fuzzy inputs to fuzzy outputs by

**Table 1 Analysis of advantages and disadvantages of genetic algorithm.**

| Literature | Literature method | Advantages | Disadvantages |
|---|---|---|---|
| *Alhijawi & Awajan (2024)* | Enhanced selection operator, fitness evaluation, and elite operation | Significantly improves the search capability of Genetic Algorithm (GA) | May increase algorithm complexity, leading to higher computational costs; sensitive to parameter settings, improper parameters may degrade performance |
| *Abdelkhalek et al. (2024)* | Adaptive adjustment of replication rate, crossover rate, and mutation rate | Consistently achieves global optimal solutions | Adaptive adjustment mechanism may increase the difficulty of algorithm implementation; in certain specific scenarios, adaptive strategies may be less effective than fixed parameter strategies |
| *Wang et al. (2024)* | Periodic encoding and priority decoding, adaptive probability strategy, multi-population joint optimization | Enhances the accuracy of GA, superior to traditional methods and GA combined with Time Series Network (TSN) | Complex implementation, may increase algorithm runtime; multi-population joint optimization may introduce additional communication overhead and synchronization challenges |
| *Usvakangas, Tuovinen & Neittaanmäki (2024)* | Greedy algorithm PMX (Partially Mapped Crossover), inversion operator, immigration strategy | Effectively addresses limitations of traditional GA | Greedy algorithm may lead to local optimum rather than global optimum; inversion operator and immigration strategy may increase algorithm complexity and runtime |
| *Mazouzi et al. (2024)* | Adaptive multi-population GA | Significantly improves efficiency, approaching or achieving optimal solutions | Adaptive mechanism may increase algorithm implementation difficulty and computational cost; the presence of multiple populations may make the algorithm difficult to implement in resource-constrained environments |
| *Pan et al. (2024)* | Refined mutation strategy in GA | Effectively reduces storage space waste, energy consumption, and delay | Refined mutation strategy may reduce algorithm exploration capability, increasing the risk of getting stuck in local optimum; in certain cases, refined strategy may be less effective than traditional mutation strategy |

constructing a fuzzy rule base and performing fuzzy reasoning, subsequently converting the fuzzy outputs into specific control signals to achieve system decision-making.

In decision-making research, uncertain decisions are generally divided into two categories: risky decisions, where decision-makers can precisely estimate the uncertainties they face (*Dekker & Alevizos, 2024*), and fuzzy decisions, where such precise estimations are not possible (*Demir, Riaz & Deveci, 2024*). During the process of expressing decision-making information, subjective ambiguities often arise. For instance, experts may find it challenging to provide a specific evaluation value when assessing a target, instead offering only a vague or imprecise range. To address this issue, the fuzzy sets introduced in *Mendel (2024)* and subsequent expansions provide an effective solution. These methods use fuzzy information representations, such as fuzzy numbers, instead of precise values to characterize the initial evaluations given by experts.

In practical decision-making scenarios, fuzzy sets are a powerful and effective tool for comprehensively addressing uncertain preference information. They enable a nuanced expression of decision-makers' uncertainties and significantly enhance the rationality and credibility of decision-making outcomes. However, fuzzy logic control (FLC) relies heavily on expert experience and the construction of fuzzy rule bases, which can lead to increased subjectivity and imprecision in the decision-making process. FLC may not provide

sufficient precision and reliability in scenarios requiring highly accurate and objective decisions. Furthermore, when confronted with vast amounts of complex and ever-changing decision-making information, maintaining and updating the fuzzy rule base can become cumbersome and time-consuming. In the specific context of this article's research, these drawbacks may limit the effectiveness and practicality of FLC. Therefore, we must explore other, more suitable methods for handling uncertainties and fuzziness in decision-making.

## METHODOLOGY

GA's robust global search capabilities enable it to effectively explore and identify potentially high-quality solutions within complex financial decision-making spaces. In this article, we can iteratively refine solutions by leveraging the natural selection and genetic mechanisms inherent in GAs until a nearly optimal resource allocation strategy is identified. Additionally, GAs can be readily integrated with fuzzy logic, further enhancing the model's predictive performance and optimization effectiveness. Consequently, adopting GAs as a model optimization method is driven by the dual considerations of efficiency and effectiveness in addressing enterprise financial management optimization problems.

This article employs a multi-objective mathematical model to clearly define the optimization objectives of financial management, such as reducing costs and improving capital utilization, thereby providing direction for the entire optimization process. Genetic algorithms are utilized to optimize the parameters and structure of neural networks, addressing the challenge of traditional methods in determining optimal parameter combinations and consequently enhancing the accuracy of predictions and decisions. The ultimately constructed HRL-FR model further incorporates fuzzy inference and hierarchical reinforcement learning to tackle uncertainties and complexities in financial management, enabling intelligent decision-making.

### Multi-objective mathematical model construction

The multi-objective mathematical model is designed to set optimization objectives such as cost reduction, capital utilization improvement, and economic benefit increase based on the enterprise's strategic planning and market environment. Specifically, cost reduction aims to decrease the enterprise's operating costs by optimizing cost structure and resource allocation. Capital utilization improvement aims to enhance the efficiency of capital usage through optimized investment strategies and capital flows. Lastly, economic benefit increases aim to boost the enterprise's profitability through comprehensive optimization.

When constructing the model, we consider various factors, including the enterprise's financial status, operating environment, market conditions, and risk tolerance. Additionally, to simplify the problem, we directly assume linear relationships among these variables to form the foundation of the multi-objective mathematical model. To achieve these objectives, the first step is to identify the decision variables that influence them. These variables may encompass the company's investment strategy, cost allocation, and capital flow. For instance, a company aiming to optimize its financial management might focus on

specific objectives like cost reduction, improved capital utilization, and increased economic efficiency.

The optimization objectives can be categorized into three main areas: cost reduction (C), increased utilization of funds (C2) and increased economic benefits (C3).

The financial costs of a firm are primarily influenced by the cost coefficients and the scale of activities undertaken. Let $c_i$ represent the cost coefficient of the $i^{\text{th}}$ activity and $x_i$ represent the number or size of the $i^{\text{th}}$ activity. Then:

$$C_1 = \sum c_i \cdot x_i. \tag{1}$$

Capital utilization reflects the efficiency with which a company uses its capital, encompassing metrics such as return on investment (ROI) and asset turnover. The objective is to maximize this function to enhance the efficiency of capital utilization and increase economic benefits. Let $K$ denote the total investment, then:

$$C_2 = \frac{K - C_1}{K} \tag{2}$$

$$C_3 = K - C_1. \tag{3}$$

## Integration of GAs and BP neural networks

Parameter determination is a critical step for the multi-objective mathematical model established above. Traditional methods for parameter determination often rely on manual expertise, which makes it challenging to identify the optimal parameter combination. By employing GA, these parameters can be automatically searched and optimized. This article integrates GA with a neural network to further enhance the optimization of enterprise financial management—encompassing tasks such as predicting financial conditions, assessing investment risks, and optimizing resource allocation.

In this approach, GA is used to optimize the neural network's parameters and structure. Through the search and optimization capabilities of GA, the parameter combination that maximizes the neural network's performance is identified, thereby improving the accuracy of predictions and decision-making. The neural network is a fitness function within the GA that evaluates the enterprise's financial performance under various parameter combinations. The specific process is illustrated in Fig. 1.

We employ BP neural networks to optimize nonlinear identification and prediction in corporate financial management. BP neural networks adjust connection weights based on established learning rules. In GAs, the population size is critical in determining their effectiveness. In this study, during the initialization of the population size (representing the number of weight combinations), the network weights are randomly assigned to the network structure. The network is then trained using a training dataset, and the inverse of the sum of squared errors is calculated to evaluate the fitness of the chromosomes.

Next, we apply crossover and mutation operations to the weights. The crossover operator randomly selects crossover points and exchanges weight values between parental chromosomes, generating offspring that inherit genetic information from both parents. The mutation operator, by contrast, randomly selects values from the initial distribution

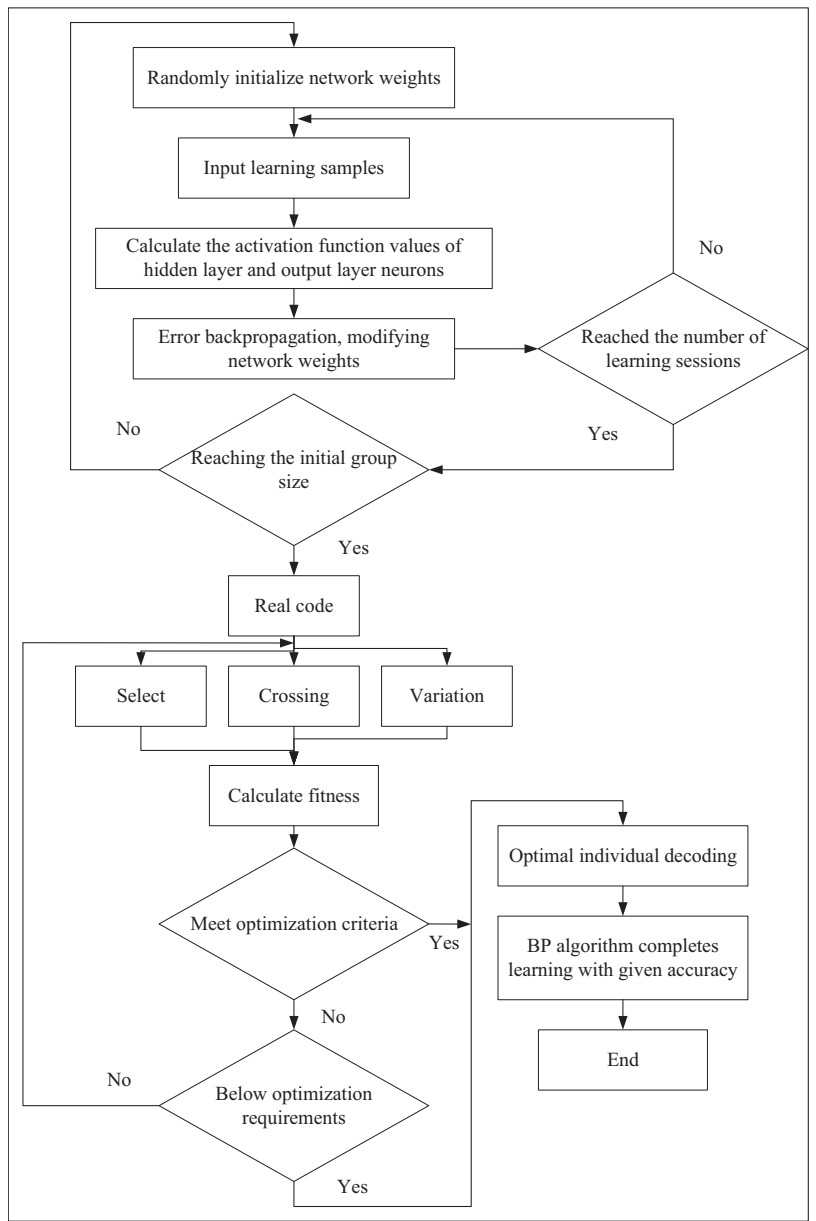

**Figure 1 The algorithm flow of GA and BP neural network.**

with a mutation probability $Pm$ and adds them to the offspring weights to introduce diversity.

After computing each individual's fitness in the population, we adopt a selection strategy to propagate individuals with higher fitness to the next generation, progressively guiding the solution toward the optimal solution space. To achieve this, the fitness proportionate selection method is used. This method ensures that the probability of an individual being selected is directly proportional to its fitness value. Specifically, individuals with higher fitness are more likely to be selected and passed on to the

next generation, enabling the algorithm to evolve consistently toward more optimal solutions.

For individuals with fitness $F_i$, the weighted individual is assigned a selection probability $P_k$:

$$P_k = \frac{F_i}{\sum_i^N F_k} \tag{4}$$

where N is the population size, we unconditionally pass on the individual with the greatest fitness to the next generation in corporate financial management optimization.

The learning is conducted in three key stages to determine neural network weights. In the first stage, the BP algorithm is employed to establish suitable initial weights for initial learning. In the second stage, GA is introduced to optimize the network learning further, bringing it closer to preset accuracy. Finally, as the GA's optimization process slows down when approaching the desired accuracy, the learning transitions back to the BP algorithm in the third stage to achieve the target accuracy. This hybrid strategy creates a network model that is stable and capable of rapid global convergence and demonstrates excellent memory and generalization capabilities.

## Hierarchical reinforcement learning based on fuzzy inference

Building on the predictive model presented in "Integration of GAs and BP Neural Networks", we leverage human prior knowledge to achieve end-to-end learning and training. To this end, we construct the HRL-FR, as illustrated in Fig. 2. This model comprises a fuzzy rule-based inference engine driven by prior knowledge and a hierarchical structure of two strategies.

The HRL-FR model generates fuzzy logic and rules aligned with corporate financial management objectives. These rules are fed into the inference engine to produce reasoning outcomes, which are input into the strategy network. This process facilitates faster model convergence, enhancing its efficiency and effectiveness.

The fuzzy rule-based reasoning machine driven by prior knowledge, as depicted in Fig. 2, incorporates multiple fuzzy rules. These rules take the current environmental state as input and, through reasoning, output the preference value for the recommended action. Taking the $i$-th rule as an example,

$$Rule\ i: IF\ \left(S_1\ is\ M_{i,1}\right)\ and\ \left(S_2\ is\ M_{i,2}\right)\ and\ \dots\ and\ \left(S_n\ is\ M_{i,n}\right),\ THEN(Action\ is\ a_i) \tag{5}$$

where $S_n$ denotes the state variables of different parts of the environment, $M_{i,n}$ denotes the fuzzy set corresponding to $S_n$, according to the fuzzy rule's arithmetic method, the conclusion of rule $i$ can be drawn, *i.e.*, to take the degree of preference of action $a_i$, and thus the preference value is the output of the rule.

Each fuzzy condition under rule i corresponds to a trainable weight value $w_i^n$, rule i also has a trainable weight value in its white body $w_i^{n+1}$, then the action preference value output by rule i is:

$$p_i = w_i^{n+1} \min\left[w_i^1 u_{i1}(s_i), w_i^2 u_{i2}(s_2), \dots, w_i^1 u_{in}(s_n)\right]. \tag{6}$$
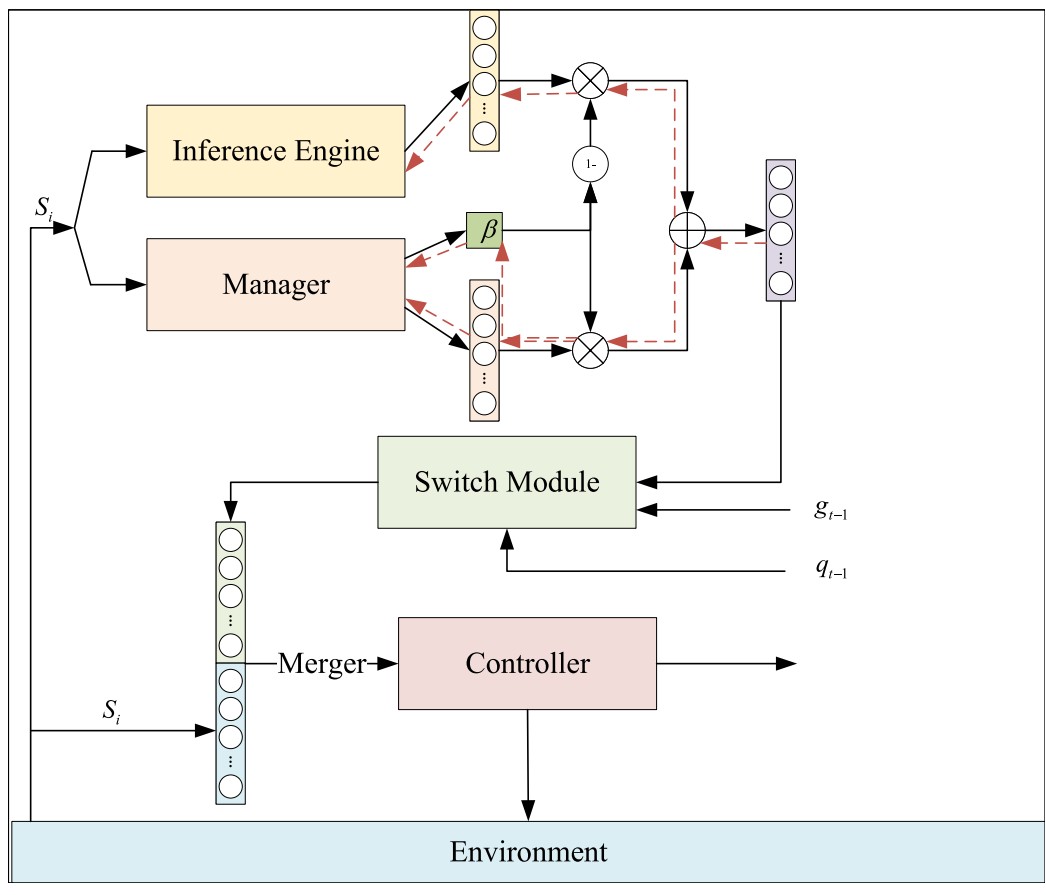

**Figure 2 The framework of HRL-FR.**

The structure of the fuzzy rule-based inference machine driven by prior knowledge is illustrated in Fig. 3. Taking the action $a_i$ of each rule as an example,

$$IF(S_1 \text{ is } M_{1,1}) \text{ and } (S_2 \text{ is } M_{1,2}) \text{ and } \ldots \text{ and } (S_n \text{ is } M_{1,n}), THEN(p_1) \text{ or} \tag{7}$$

$\ldots$

$$IF(S_1 \text{ is } M_{i,1}) \text{ and } (S_2 \text{ is } M_{i,2}) \text{ and } \ldots \text{ and } (S_n \text{ is } M_{i,n}), THEN(p_i) \text{ or} \tag{8}$$

$\ldots$

$$IF(S_1 \text{ is } M_{m,1}) \text{ and } (S_2 \text{ is } M_{m,2}) \text{ and } \ldots \text{ and } (S_n \text{ is } M_{m,n}), THEN(p_m). \tag{9}$$

Each rule computes the preference value for its corresponding action for a discrete action space. When multiple rules correspond to the same action, they are combined using the logical "or" operation, which is implemented through the maximization operator. This operation consolidates the preference values to form the resultant set and determines the action $a_i$ to be taken. The preference value for taking a specific action is denoted as $p_{a,i}$. By organizing the preference values of all possible actions, a preference vector for all actions is constructed:

$$P = [p_{a,1}, p_{a,2}, \ldots, p_{a,m}]. \tag{10}$$

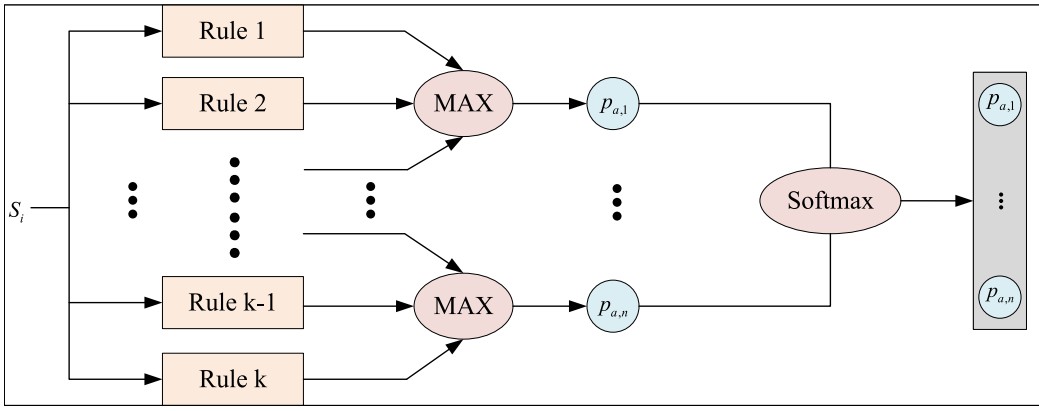

**Figure 3 Fuzzy rule inference machine based on prior knowledge.**

Next we combine the fuzzy rule inference machine with the upper level controller. The final action decision $g_t'$ of the upper-level manager is

$$g_t' = b \cdot \widetilde{g_{t,1}} + (1 - b)\widetilde{g_{t,2}} \tag{11}$$

where $\widetilde{g_{t,1}}$ and $\widetilde{g_{t,2}}$ all undergo a sigmoid operation. The weight value $b$ is the one-dimensional output of the manager, so this parameter is also trainable.

In the HRL-FR model, in order to utilize the prior knowledge, we use the rough policy given by the fuzzy rule inference module to guide the agent in interacting with the environment at an early stage. As shown in Fig. 1, the upper-level manager receives the environment states and outputs an action decision $\widetilde{g_{t,1}}$ and a weight value $b$, the reasoner receives the environment states and outputs the reasoned preference action $\widetilde{g_{t,2}}$, and $g_t'$ represents the final preference subgoal vector.

The intermediate goal $g_t'$ is input into the switching module, where it is combined with the switching signal output from the controller at the previous moment and the subgoal $g_{t-1}$ from the previous moment to generate the final subgoal $g_t$. This subgoal is concatenated with the external environmental state $s$ and input into the lower-level controller's policy network. The controller subsequently outputs the action $a$ for the intelligent agent to interact with the environment.

In this process, the weights in the HRL-FR model are represented as one-dimensional trainable variables dynamically output by the lower-level strategy in real time. These weights adjust in response to changes in the external environmental state, making the model more adaptable and better aligned with the actual decision-making needs of enterprise financial management.

## EXPERIMENTAL ANALYSIS

In this section, we evaluate the performance of the proposed enterprise financial management optimization model by comparing it with existing financial risk management prediction methods. Additionally, we analyze the performance of the integrated GA and

BP neural network model and the effectiveness of intelligent decision-making facilitated by fuzzy reasoning.

## Experimental data

This study utilizes two prominent datasets, Compustat and Center for Research in Security Prices (CRSP), for its experimental analysis in enterprise financial management optimization modeling. Compustat primarily covers annual and quarterly financial data for most publicly traded companies in North America. It encompasses key financial statements such as income, balance sheets, and cash flow statements. Additionally, it includes company basic information, shareholder structure, and other pertinent details. Compustat typically spans decades of data, providing a comprehensive historical perspective invaluable for financial analysis and modeling. In the study, Compustat was used to analyze various financial indicators such as the working capital asset ratio, debt-to-equity ratio, and others. These indicators were found to significantly impact the prediction results, validating the effectiveness of the proposed model. The dataset was also instrumental in training and testing the fusion of genetic algorithms and BP neural networks for financial risk prediction.

CRSP includes basic information such as prices, returns, and trading volumes. Additionally, it offers a wealth of data on market indices, mutual funds, and other investment vehicles. Due to its comprehensive nature, CRSP is widely used in financial research to analyze market trends, performance, and risk. CRSP was utilized alongside Compustat to provide a broader view of the financial market. The dataset was used to evaluate the performance of the proposed HRL-FR in predicting and identifying enterprises' future financial management information.

## Experimental evaluation criteria

In prediction modeling, risk identification error rate, misclassification rate, and prediction accuracy rate are essential metrics for assessing model performance.

The margin of error is commonly used to measure the degree of difference between the predicted or estimated value and the true value. It is calculated by the Formula (12):

$$E = \left( \frac{|R - P|}{R} \right) \times 100\% \tag{12}$$

where $R$ is the target value to be predicted or estimated, $P$ is the predicted value.

The misclassification rate is the ratio of the number of samples incorrectly predicted by the model to the total number of samples. For binary classification problems, the misclassification rate can be further subdivided into CTW (Correct to Wrong) and WTC (Wrong to Correct). CTW refers to the number of samples with an original label of 1 (positive samples) but predicted as 0 (negative samples). WTC refers to the number of samples with an original label of 0 (negative samples) but predicted as 1 (positive samples):

$$ER = \left( \frac{CTW + WTC}{Num} \right) \times 100\% \tag{13}$$

where $Num$ is the total sample size.

Predictive accuracy is the probability of a sample being positive out of all samples predicted to be positive. It is calculated by the Formula (14):

$$Pre = \left( \frac{TP}{TP + FP} \right) \times 100\%. \tag{14}$$

TP is the number of samples the classifier correctly predicted as positive cases. fP is the number of samples the classifier incorrectly predicted as positive cases.

## Financial management risk forecasting

The BP neural network employed for corporate financial management modeling consists of an input layer with five nodes, a first hidden layer with 20 nodes, a second with eight nodes, and an output layer with four nodes. The BP neural network learning rate is 0.001. In the GA used for optimization, choosing a population size of 10 is specifically tailored for scenarios with limited computational resources or more minor problem scales. Setting the mutation probability to 0.002 means that the algorithm will randomly alter the genes of individuals with a small probability in each iteration, which helps introduce new gene combinations and prevents premature algorithm convergence. Furthermore, the following parameters are applied: crossover probability $P = 0.5$, fitness function $T = 1/(1 + e)$, where $e = 0.1$, and the number of generations for genetic iterations $g = 50$.

Genetic operations are performed using these configurations on the individuals in the population. Subsequently, a comparative study method is employed to analyze the prediction model's performance for managing financial information, integrating the GA and BP neural networks.

(1) On the CRSP dataset, (2) on the Compustat dataset

To avoid multicollinearity in financial management forecasting, multicollinearity detection is conducted on the forecasting model integrating the GA and BP neural networks. The analysis evaluates eight key indicators related to corporate financial management optimization: working capital asset ratio ($x1$), debt-to-equity ratio ($x2$), long-term debt-to-equity ratio ($x3$), accounts receivable turnover ratio ($x4$), return on net assets ratio ($x5$), five-loss ratio ($x6$), growth rate of primary business operations ($x7$), and expansion rate of total assets ($x8$).

The results in Fig. 4 demonstrate that the tolerance (TOF) values for all variables are less than 1, indicating no significant multicollinearity among the variables in the prediction model. Furthermore, all variables' variance inflation factor (VIF) values are relatively low, confirming that these eight indicators are critical factors influencing enterprise financial management.

(1) Optimization of variable coefficients

(2) T-test

(3) P-test

(4) The correlation.

We compare the LPM risk prediction model (*Kalra et al., 2024*) with the financial information management risk prediction model that integrates GA and BP neural

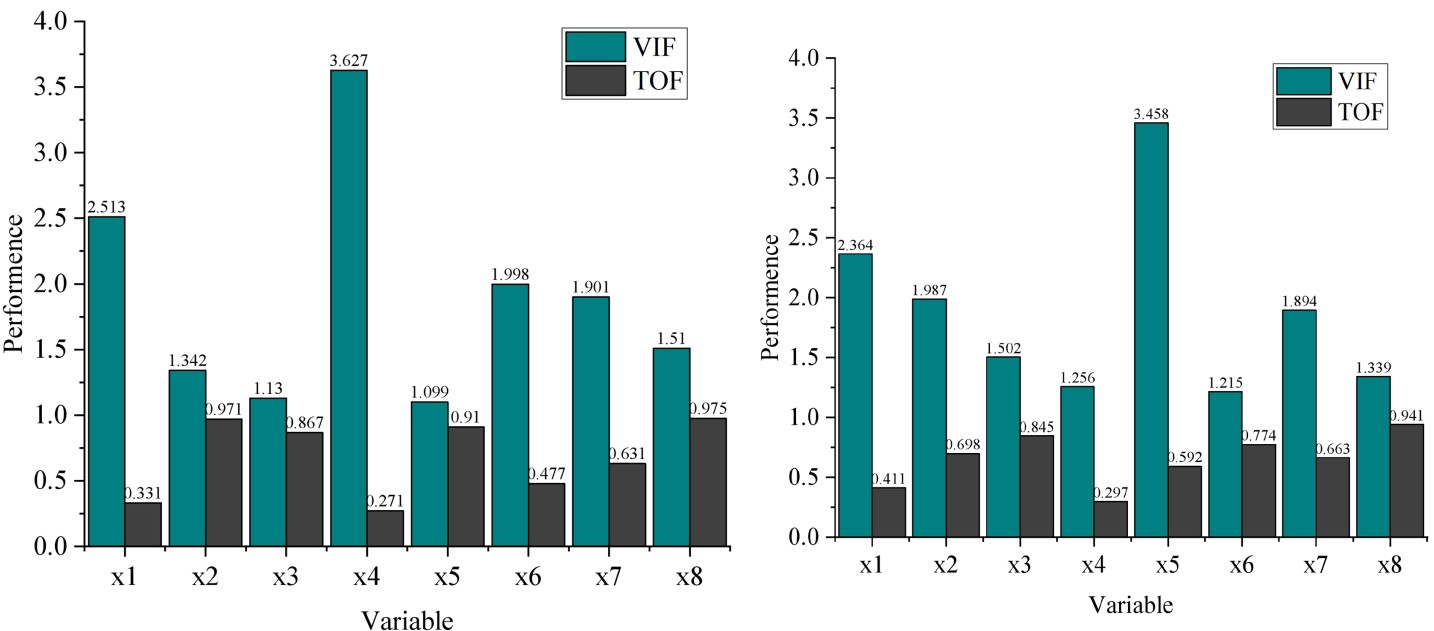

**Figure 4  Optimal ethical decision tree.**

networks using the Compustat dataset, as illustrated in Fig. 5. The comparative analysis of the prediction results reveals that the regression outcomes of the financial information management risk prediction model incorporating GA and BP neural networks fare closer to the maximum likelihood value, demonstrating superior predictive performance.

Regarding parameter optimization, analysis of sub-figure (1) alongside other sub-figures shows that the coefficient for the working capital asset ratio ($X1$) in this model is 0.325, with a $P$-value of less than 0.001, indicating a significant positive impact on the prediction results. Similarly, the debt-to-equity ratio ($X2$) and the main business growth rate ($X7$) also exhibit significant positive effects on the prediction results.

Conversely, the coefficients of the long-term debt-to-equity ratio ($X3$) and return on net assets ($X5$) are negative, signifying significant negative effects on the prediction results. Notably, while the accounts receivable turnover ratio ($X4$) has a positive coefficient, its t-value is negative, and its $P$-value is less than 0.001, indicating a significant negative correlation with the model's prediction outcomes.

Overall, the integration of GA effectively optimizes the parameters, leading to a better fit of the BP neural network model and higher accuracy in financial risk prediction.

The specific risk prediction results are presented in Fig. 6. On the Compustat dataset, the proposed prediction model demonstrates a clear downward trend in the risk misclassification rate, decreasing steadily from a relatively high level in 2016 to a significantly lower level in 2022. In contrast, the risk misclassification rate of the LPM model exhibits noticeable fluctuations. Although it also shows a general downward trend, its variability is more significant, and it never achieves the lower levels reached by the prediction model proposed in this article.

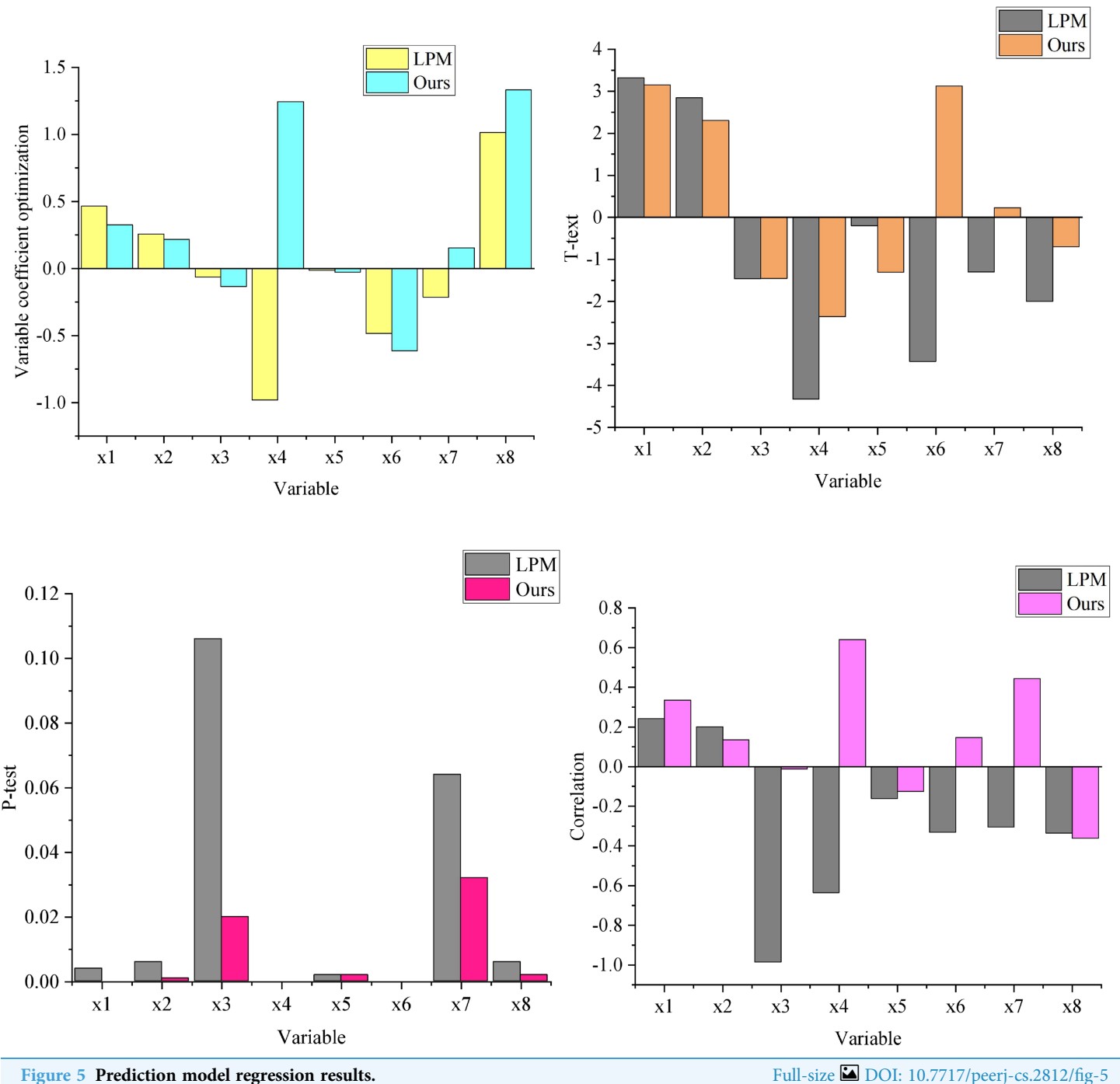

**Figure 5 Prediction model regression results.**

Notably, from 2016 to 2022, the risk misclassification rate of the proposed model consistently remains lower than that of the LPM model, a distinct advantage reflected in the figure. Further analysis reveals that the risk prediction accuracy of the proposed model follows a trend of initial decline, followed by improvement. Specifically, the accuracy rate decreased between 2016 and 2018 but rose steadily after 2019, reaching a relatively high

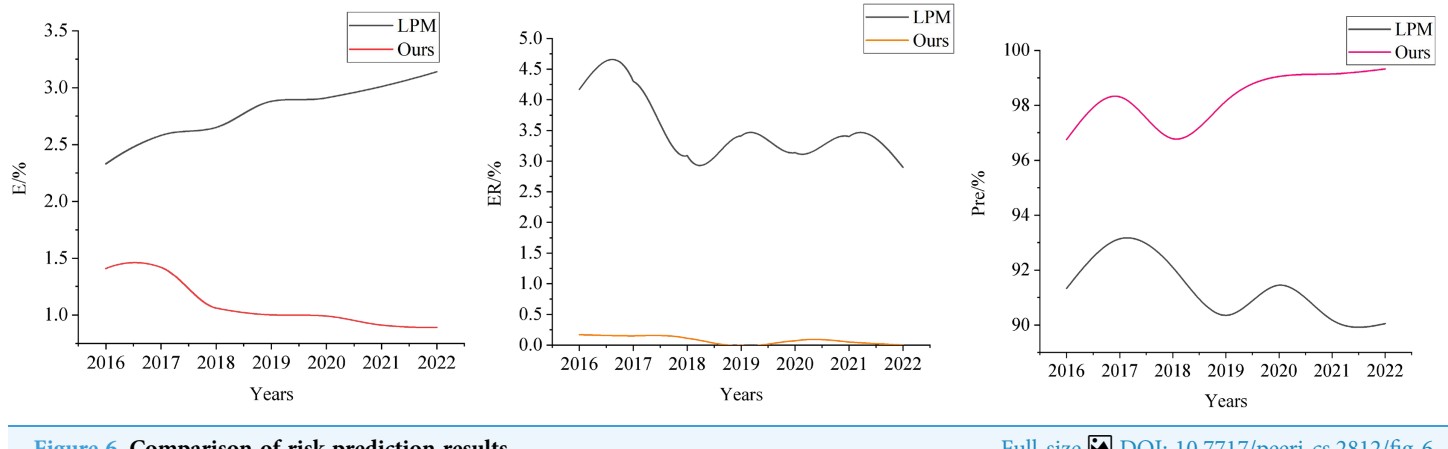

**Figure 6 Comparison of risk prediction results.**                               

level in 2022. In contrast, the LPM model maintains a relatively low accuracy level throughout the same period, with considerable fluctuations.

Our genetic algorithm is used to optimize the parameters and structure of the neural network, enhancing the model's prediction capabilities. The BP neural network handles nonlinear identification and prediction in financial management. The HRL-FR model, through hierarchical decision-making and strategy optimization, combined with fuzzy reasoning to deal with uncertainty, enables reasonable decision-making in complex and dynamic environments. Furthermore, we carefully selected key variables and eliminated those without significant impact on the prediction results through multicollinearity detection, improving the model's accuracy. Finally, the risk prediction accuracy of the proposed model is significantly higher than that of the LPM model in every year analyzed. This advantage becomes particularly pronounced after 2019, underscoring the superior performance and reliability of the proposed model in financial risk prediction.

## Performance analysis of intelligent decision making

Finally, a comprehensive assessment is conducted using sample data from the CRSP and Compustat datasets to evaluate the performance of the enterprise financial management optimization model in terms of gainfulness and comprehensive profit; the results are shown in Fig. 7. We compare two decision algorithms, DS-ID (*Waqar, 2024*) and DRL-ID (*Ju & Zhu, 2024*; *Xia et al., 2023*; *Zhao et al., 2024*; *Chang, Gao & Li, 2025*).

When comparing the HRL-FR model with the two decision-making algorithms, DS-ID and DRL-ID, we can observe certain limitations in the latter two. While the DS-ID algorithm provides intelligent decision support, it may overly rely on specific engineering domain knowledge, leading to limited generalization ability in the complex and ever-changing financial management environment. This makes it difficult to comprehensively capture subtle changes and potential risks in corporate financial data. On the other hand, although the DRL-ID algorithm assesses corporate financial asset risks and makes intelligent decisions based on reinforcement learning, it may suffer from the inherent

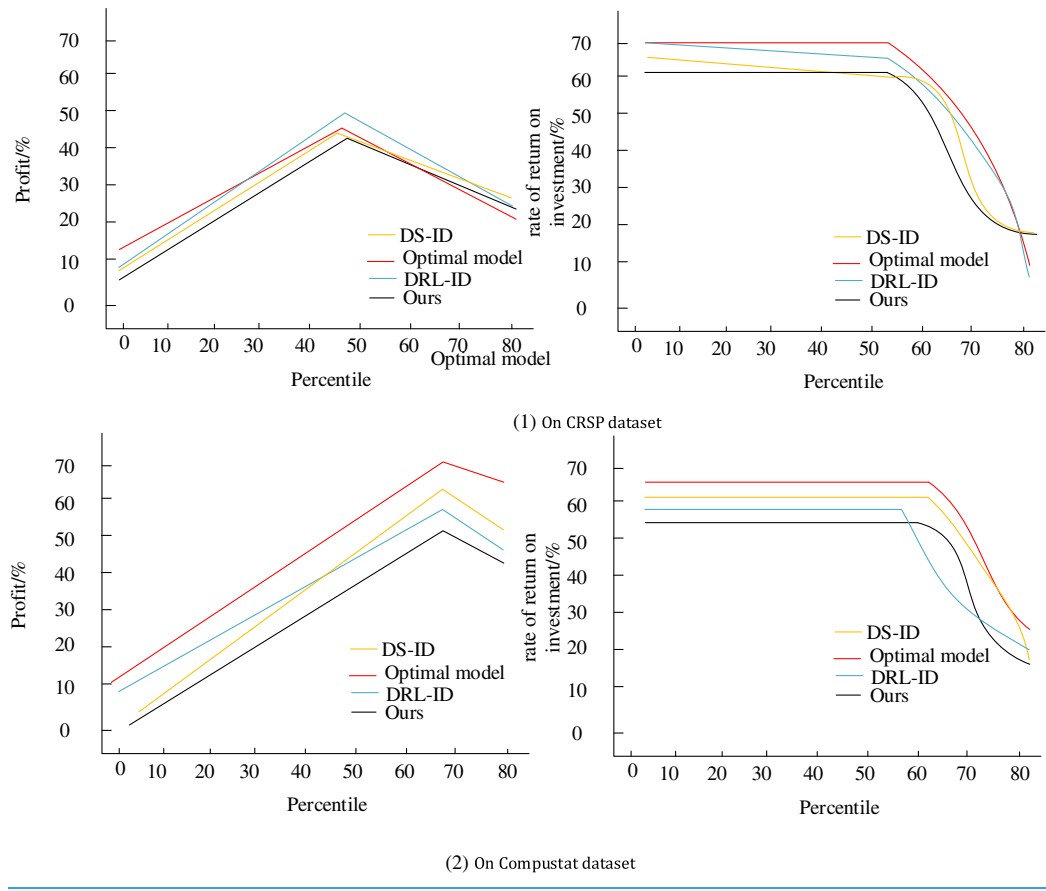

**Figure 7 Intelligent decision performance analysis.**

exploration-exploitation dilemma of reinforcement learning. When confronted with highly uncertain and dynamically changing financial management scenarios, its decision-making efficiency and stability need improvement. In contrast, the HRL-FR model, through hierarchical decision-making and policy optimization combined with fuzzy reasoning to handle uncertainty, can more flexibly adapt to the complexity and variability of the financial management environment. As a result, it demonstrates excellent performance in terms of profitability and overall revenue.

The fitted curve plots align closely with those of the optimal model, visually showcasing the HRL-FR model's superior gainfulness and comprehensive profit performance. In practical scenarios, enterprises can integrate the model into their financial management systems to achieve automated and intelligent decision-making processes. Furthermore, the multi-objective nature of the model helps enterprises balance the relationships between different economic indicators, maximizing overall benefits. Enterprises can adapt to the ever-changing market environment by continuously optimizing and improving the model, enhancing their competitiveness and profitability.

In exploring the application of the combination of genetic algorithms and neural networks in financial management, this article proposes innovative and innovative methods. We have not only designed a completely new optimization algorithm to enhance

the efficiency and accuracy of the model in processing complex financial data but also innovatively introduced fuzzy logic into this integrated model to bolster its intelligent decision-making capabilities in uncertain environments. This comprehensive approach not only enriches the theoretical framework of the integration of genetic algorithms and neural networks but also provides unprecedented intelligent solutions for enterprise financial management practices, demonstrating this article's unique contribution and academic value.

## CONCLUSION

This article constructs a multi-objective mathematical model, integrating GAs and neural networks to optimize parameters for cost reduction, improved capital utilization, and increased economic benefits. This approach addresses the limitations of traditional methods in determining optimal parameter combinations, significantly enhancing prediction accuracy and decision-making capabilities. Furthermore, the HRL-FR model accelerates convergence and improves decision-making performance. Despite the outstanding performance of the proposed HRL-FR model in optimizing corporate financial management, it still has some limitations. For example, when faced with extremely complex or large-scale data sets, the model's computational efficiency and convergence speed may be affected, leading to a decline in performance. Additionally, when there are severe conflicts between different objectives, the model may require more trade-offs and compromises, which may affect its practicality and effectiveness. Therefore, it is necessary to fully consider these factors to optimize the model's performance and practicality in practical applications.

To practically implement the HRL-FR model in a corporate environment, it is essential to consider multiple aspects comprehensively, including regulatory compliance, system integration, and interpretability. By collaborating closely with partners, legal experts, and internal teams, businesses can overcome these challenges and fully harness the potential of the HRL-FR model to optimize their financial decisions and risk management strategies. Future research will focus on improving algorithms to enhance computational efficiency and convergence speed. Efforts will also be directed toward exploring broader application scenarios to validate and optimize the model's practicality and universality. Moreover, an in-depth investigation into the fusion mechanism of GAs and fuzzy logic will aim to provide more efficient and accurate optimization solutions for enterprise financial management.

### Funding

This study was supported by the Youth Scientific Research Fund of Zhengzhou University of Economics and Business (Fund number: QK2403). The funders had no role in study design, data collection and analysis, decision to publish, or preparation of the manuscript.

## Grant Disclosures

The following grant information was disclosed by the authors:
Youth Scientific Research Fund of Zhengzhou University of Economics and Business:
QK2403.

## Competing Interests

The authors declare that they have no competing interests.

## Author Contributions

- Sujuan Wang conceived and designed the experiments, performed the experiments, analyzed the data, performed the computation work, prepared figures and/or tables, authored or reviewed drafts of the article, and approved the final draft.
- Musadaq Mansoor conceived and designed the experiments, performed the experiments, analyzed the data, authored or reviewed drafts of the article, and approved the final draft.

## Data Availability

The CRSP data is available at Zenodo: Nitiraj Kulkarni, & Jagadish Tawade. (2024). Dataset: CRISPR Therapeutics AG (CRSP) Stock Performance [Data set]. Zenodo. https://doi.org/10.5281/zenodo.12555376

The Compustat Financials dataset is available at https://www.marketplace.spglobal.com/en/datasets/compustat-financials-(8).

## Supplemental Information

Supplemental information for this article can be found online at http://dx.doi.org/10.7717/peerj-cs.2812#supplemental-information.

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
