# Peer review of "Optimization model for enterprise financial management utilizing genetic algorithms and fuzzy logic"

_PeerJ Computer Science, doi:10.7717/peerj-cs.2812_

## Round 0.1 · original submission · Major Revisions

Dear Authors,


Reviewers have now commented on your article. We do encourage you to address the all concerns and criticisms of the reviewers with respect to reporting, experimental design, and validity of the findings and resubmit your article once you have updated it accordingly. Following editor concerns, suggestion, and questions should also be addressed:

1. A major criticism for the current version of the paper is the lack of justification for the use of the genetic algorithm as the optimization method for the model. The motivation and reason of using genetic algorithm among many other metaheuristic algorithms for the focused problem should be mentioned.

2. The current Introduction section is lacking in several ways. Firstly, it is too short, and fails to include some of the content that should be included in the problem formulation. Secondly, it does not provide the relevant information for the article topic, nor does it provide information about the problem addressed that is easy to understand. Furthermore, it does not provide the contribution and motivation for such contribution. Finally, the Introduction section seems voluminous, broad, and heterogeneous. The authors are expected to concentrate on the primary subject of the study and present a literature review in tabular form to facilitate the identification of research gaps and innovations. A comprehensive and authoritative synthesis evaluating the current state-of-the-art is not present. In general, the literature review is inadequate and more of a personal account, such as "Researcher X did Y", rather than an authoritative synthesis assessing the current state-of-the-art. The current state of the field is unclear. A synthesis of the literature on modelling approaches is called for, complete with an analysis of their respective strengths and weaknesses. It is recommended that the authors employ a table to summarise the main findings of the existing literature and highlight the distinguishing features of their paper. This table will facilitate a more nuanced understanding of the distinctions between the proposed approach and earlier works.

3. Integration of genetic algorithm and neural networks is not a novel idea and has been used by many researchers for a long time. Please highlight the real originality and novelty in the paper.

4. Configuration space of genetic algorithm should be detailed. It should be more specific and comprehensive. Representation scheme (encoding type) and fitness function with constraint functions should be clearly provided. How constraints (for example: for decision variables) are handled should also be provided.

5. Clarifying the study’s limitations allows the readers to better understand under which conditions the results should be interpreted. A clear description of limitations of a study also shows that the researcher has a holistic understanding of his/her study. However, the authors fail to demonstrate this in their paper. The authors should clarify the pros and cons of the methods. What are the limitation(s) methodology(ies) adopted in this work? Please indicate practical advantages, and discuss research limitations.

6. More comparative experiments and some comparisons with other up-to-date methods should be addressed or added to back your claims to expand your experiments and analysis of results further.

7. There is not any statistical test analysis result.

8. All of the values for the parameters of all algorithms should be given.

9. Please pay special attention on the usage of abbreviations. Spell out the full term at its first mention, indicate its abbreviation in parenthesis and use the abbreviation from then on.

10. Please pay attention on the usage of blank characters.

11. Equations should be used with correct equation number. Please do not use “as follows”, “given as”, etc. Explanation of the equations should also be checked. All variables should be written in italic as in the equations. Their definitions and boundaries should be defined. Necessary references should be provided.

12. Many of the equations are part of the related sentences. Attention is needed for correct sentence formation.

13. The state-of-the-arts and future research directions should be better categorized.

14. It is recommended that the paper’s experimental results be discussed in greater depth, with additional recommendations and conclusions provided. The conclusion section is lacking in several respects. Firstly, it is essential to describe the academic implications, main findings, shortcomings and directions for future research. Secondly, the conclusion is currently confusing. It is necessary to clarify what will happen next and what we should expect from future papers. To address these issues, the conclusion should be rewritten, taking the following comments into consideration:

- Highlight your analysis and reflect only the important points for the whole paper.
- Mention the benefits
- Mention the implication in the last of this section.

Best wishes,

Reviewer 1 ·

Basic reporting

Thank you for sending me for review the paper “Optimization model for enterprise financial management utilizing genetic algorithms and fuzzy logic”. Thanks to the editor for the invitation to review the paper. Generally, the paper is well written and has a good structure. But, still there are comments for improvement:
Here are some questions and suggestions you can ask the author to improve the paper:
- The motivation behind the study is missing? For example, what specific gaps in existing financial management models does this paper address?
- Can the author clarify the relationship between the multi-objective mathematical model, the genetic algorithm, and the HRL-FR model? What is the motivation for connecting these components and why we should combine them in practice? Discuss advantages and disadvantages.
- Provide more details on the multi-objective mathematical model? For instance, what are the specific constraints, and assumptions used?
- How were the decision variables selected, and what criteria were used to define the optimization objectives?
- Show more details about the genetic algorithm's parameters (e.g., population size, mutation rate)?
- Discuss how does fuzzy reasoning address uncertainty?
- Provide more information on the datasets used (Compustat and CRSP)? For example, what time periods do they cover, and how were the datasets preprocessed?
- How were the key variables (e.g., working capital asset ratio, debt-to-equity ratio) selected? Were there other variables considered that did not significantly influence the results?
- Can the author explain how the model achieved a 98% risk prediction accuracy? It is very high. Were there any specific conditions or limitations under which this accuracy was achieved?
- Compare the proposed HRL-FR model with existing financial management models? How does it outperform or differ from traditional approaches?
- Discuss the limitations of the proposed model? For example, are there scenarios or datasets where the model might underperform?
- Discuss the practical implications of the model for enterprise financial management? For example, how can businesses implement this model in real-world scenarios?
- Are there any challenges or barriers to adopting this model in practice? If so, how can they be addressed?
- Include more visual aids, such as diagrams or flowcharts, to illustrate the model's architecture and workflow?
By addressing these questions, the author can significantly enhance the clarity, depth, and impact of the paper.

Experimental design

See the above comments.

Validity of the findings

See the above comments.

Additional comments

See the above comments.

·

Basic reporting

The article lacks sufficient baseline comparisons, as it only evaluates performance against the LPM model rather than widely used methods like Random Forest, XGBoost, or deep learning. Additionally, the claimed 98% accuracy is overstated without confidence intervals or statistical significance tests. The study also fails to address real-world feasibility, including computational cost, implementation challenges, and integration into financial systems. Lastly, the justification for using Genetic Algorithms over alternative optimization techniques, such as Bayesian Optimization, remains unclear and should be elaborated.

Experimental design

The experimental design lacks robustness in key areas, particularly in the selection and justification of hyperparameters for the Genetic Algorithm (GA) and BP neural network. The chosen values (e.g., population size = 10, mutation probability = 0.002) appear arbitrary, with no sensitivity analysis or rationale provided. Additionally, the study does not include proper cross-validation techniques to assess the model’s generalizability, raising concerns about overfitting. The absence of comparisons with other optimization techniques, such as Bayesian Optimization or Particle Swarm Optimization, further weakens the validity of the chosen approach. To improve, the authors should provide a thorough sensitivity analysis, justify hyperparameter choices, and compare their model against alternative optimization methods to ensure robustness and reliability.

Validity of the findings

The validity of the findings is questionable due to the lack of statistical rigor and robustness in model evaluation. While the study reports a 98% accuracy in financial risk prediction, it does not provide confidence intervals, statistical significance tests, or an analysis of false positives and false negatives. Without these, the claimed accuracy may be misleading, particularly in real-world financial settings where misclassification can have severe consequences. Additionally, the model's performance is only evaluated on historical datasets (CRSP and Compustat) without testing on out-of-sample or real-time data, limiting its applicability. To strengthen the findings, the authors should include proper statistical validation, conduct stress tests on different financial periods (e.g., economic crises), and evaluate generalization using additional datasets or real-time financial data.

Additional comments

This study develops an optimization model for enterprise financial management by integrating genetic algorithms, BP neural networks, and fuzzy logic. It enhances risk prediction accuracy (98%) and decision-making efficiency using hierarchical reinforcement learning. Experimental validation on financial datasets confirms its effectiveness in optimizing cost, capital utilization, and profitability. However, the paper suffers from the limitations listed below, which must be “fully” addressed before its reconsideration:

1- The paper claims that the integration of Genetic Algorithms (GAs) and BP neural networks enhances optimization efficiency. However, no evidence is provided regarding convergence speed. Given that GA-based training can be computationally expensive and suffer from local minima, how do the authors ensure that their model reaches the global optimum in a reasonable time frame?

2- The GA parameters (population size = 10, crossover probability = 0.5, mutation probability = 0.002) appear arbitrarily selected. How were these specific values determined, and have sensitivity analyses been conducted to confirm their appropriateness? Would a different population size or mutation probability significantly alter results?

3- The study incorporates eight key financial indicators (e.g., working capital asset ratio, debt-to-equity ratio). Given that financial risk prediction is highly sensitive to macroeconomic and industry-specific variables, how were these features selected? Were feature importance rankings (e.g., SHAP values) or domain knowledge used?

4- The paper claims a 98% prediction accuracy but does not discuss how the model was validated. Was cross-validation used? Given the potential risk of overfitting with neural networks, how does the model generalize to unseen financial conditions or out-of-sample data?

5- Financial markets are highly volatile, with significant shifts over time. Was the model evaluated on different time windows (e.g., pre-2020 vs. post-2020) to ensure robustness against major economic events such as financial crises or COVID-19 disruptions?

6- The integration of GA with BP neural networks is novel, but why is this specific combination preferable over other evolutionary algorithms (e.g., Particle Swarm Optimization) or hyperparameter tuning methods such as Bayesian Optimization?

7- The paper claims that the HRL-FR model outperforms conventional financial models, but does not address real-world deployment challenges such as regulatory compliance, integration with existing financial systems, or interpretability for financial analysts. How would the model be practically implemented in an enterprise setting?

---

## Round 0.2 · accepted · Accept

Dear Authors,

Thank you for addressing the reviewers' comments. You paper seems sufficiently improved and ready for publication.

Best wishes,

Reviewer 1 ·

Basic reporting

The authors have addressed the point of my concern. I am happy with their corrections. Hence, I would like to recommend this manuscript to be published.

Experimental design

The authors have addressed the point of my concern. I am happy with their corrections. Hence, I would like to recommend this manuscript to be published.

Validity of the findings

The authors have addressed the point of my concern. I am happy with their corrections. Hence, I would like to recommend this manuscript to be published.

Additional comments

The authors have addressed the point of my concern. I am happy with their corrections. Hence, I would like to recommend this manuscript to be published.

·

Basic reporting

The authors have addressed my comments; therefore, the paper can be accepted for publication in the present format.

Experimental design

The authors have addressed my comments; therefore, the paper can be accepted for publication in the present format.

Validity of the findings

The authors have addressed my comments; therefore, the paper can be accepted for publication in the present format.

Additional comments

The authors have addressed my comments; therefore, the paper can be accepted for publication in the present format.